# Nutrients to Improve Mitochondrial Function to Reduce Brain Energy Deficit and Oxidative Stress in Migraine

**DOI:** 10.3390/nu13124433

**Published:** 2021-12-10

**Authors:** Michal Fila, Cezary Chojnacki, Jan Chojnacki, Janusz Blasiak

**Affiliations:** 1Department of Developmental Neurology and Epileptology, Polish Mother’s Memorial Hospital Research Institute, 93-338 Lodz, Poland; michal.fila@iczmp.edu.pl; 2Department of Clinical Nutrition and Gastroenterological Diagnostics, Medical University of Lodz, 90-647 Lodz, Poland; cezary.chojnacki@umed.lodz.pl (C.C.), jan.chojnacki@umed.lodz.pl (J.C.); 3Department of Molecular Genetics, Faculty of Biology and Environmental Protection, University of Lodz, 90-236 Lodz, Poland

**Keywords:** migraine, dietary intervention in migraine, mitochondria, energy production, glycolysis, oxidative stress

## Abstract

The mechanisms of migraine pathogenesis are not completely clear, but ^31^P-nuclear magnetic resonance studies revealed brain energy deficit in migraineurs. As glycolysis is the main process of energy production in the brain, mitochondria may play an important role in migraine pathogenesis. Nutrition is an important aspect of migraine pathogenesis, as many migraineurs report food-related products as migraine triggers. Apart from approved anti-migraine drugs, many vitamins and supplements are considered in migraine prevention and therapy, but without strong supportive evidence. In this review, we summarize and update information about nutrients that may be important for mitochondrial functions, energy production, oxidative stress, and that are related to migraine. Additionally, we present a brief overview of caffeine and alcohol, as they are often reported to have ambiguous effects in migraineurs. The nutrients that can be considered to supplement the diet to prevent and/or ameliorate migraine are riboflavin, thiamine, magnesium ions, niacin, carnitine, coenzyme Q10, melatonin, lipoic acid, pyridoxine, folate, and cobalamin. They can supplement a normal, healthy diet, which should be adjusted to individual needs determined mainly by the physiological constitution of an organism. The intake of caffeine and alcohol should be fine-tuned to the history of their use, as withdrawal of these agents in regular users may become a migraine trigger.

## 1. Introduction

Migraine is a common neurological disease with a high prevalence in people younger than 50, causing serious disability in individuals, and a significant burden for societies. It is featured by paroxysmal attacks with head pain, nausea, and increased sensitivity to light, sound, and movement [1].

Migraine may be induced by a wide spectrum of triggers, including intake or withdrawal of specific food, physical exercise, weather changes, and others. However, it is not clear why some people develop migraine in response to such triggers while others do not. Moreover, some substances or factors may act as migraine triggers when they are present, whereas others—when they are absent, e.g., coffee or, more generally, caffeine [2]. This impedes preventive strategies in migraine.

Oxidative stress is reported to associate with migraine [3]. However, there are at least two main problems when addressing this issue. Firstly, it is difficult to find a human disorder which would not include oxidative stress in its pathogenesis. Secondly, almost all, if not all, known or putative migraine triggers may be linked with oxidative stress (reviewed in [3]).

Women have, on average, three times higher risk of having migraine than men, suggesting that mitochondrial transmission may play a role in familial cases of this disease [4]. Moreover, migraine frequently associates with mitochondrial disorders [5]. Therefore, mitochondria may be an important element in migraine pathogenesis [6]. In the brain, mitochondria are energy producers using mainly glucose and oxygen as substrates for ATP production [7]. Migraine was originally considered as a hypoglycemic disorder [8]. The application of phosphorus nuclear magnetic resonance (^31^P-NMR) spectroscopy revealed changes in energy metabolism in the brain of migraineurs [9,10,11,12,13]. This suggests an important role of imbalance in brain energy demand and mitochondrial ATP production in migraine.

In summary, a triad of brain energy deficit–mitochondria–oxidative stress can be considered as a potential axis in pathways of migraine pathogenesis (reviewed in [14]). Mitochondria can be central for this axis, as they are the main energy producers in the brain and mitochondrial electron chain (ETC), essential for ATP synthesis, and produce reactive oxygen species (ROS) even in their normal functioning. These ROS can be overproduced by impaired mitochondria, contributing to oxidative stress.

The general strategy to prevent migraine attack is to “avoid triggers”, i.e., to avoid or escape migraine-supporting conditions. The diet has a high potential to prevent migraine and moderate consequences of migraine attacks, and an avoidance diet, eliminating potential migraine triggers, is recognized as an efficient prevention in many cases [15]. It should be underlined that an elimination diet does not replace anti-migraine drugs, but it may increase their efficacy. Besides elimination diets, the supplemented diet, i.e., a diet that is supplemented with nutrients, is emergingly recognized and appreciated in migraine prevention [16]. The “mitochondrial diet” (“mito diet”), a diet with nutrients to restore impaired mitochondrial functions, is considered potentially beneficial in several human illnesses [17].

In this review, we develop the thesis that impaired mitochondrial functions may contribute to migraine pathogenesis through brain energy deficit, and oxidative stress may be ameliorated by some nutrients included in the diet. Obviously, such supplementation with nutrients will have a greater potential impact when applied to a migraine elimination diet compared to a regular diet, which may contain migraine triggers. We also briefly present a potential role of caffeine and alcohol in migraine pathogenesis and prevention. The main aim of this paper is to present rationale on the involvement of mitochondria, bioenergetics, and oxygen metabolism in migraine pathogenesis in the context of the beneficial potential of nutritional intervention. We concentrate on the mechanistic aspect, rather than specific dietary recommendations and guidelines for use that can be found elsewhere (e.g., [18]). Due to the need to keep this review concise, we do not provide general information on migraine, unless closely related to the main subject. Readers interested in theories of migraine pathogenesis, contemporary therapies, diet in migraine prevention, migraine’s place among other pain-related syndromes, and other migraine-related issues are referred to other reviews, e.g., [19,20,21,22,23].

## 2. Increased Demand and Decreased Production of Energy in Migraine

Brain energy deficit between attacks in migraineurs is underlined by increased energy demand by a hyperexcitable brain, or a decreased supply of energy caused by mitochondrial impairment (reviewed in [14]).

The brain has an extremely high metabolic activity associated with a continuous supply of oxygen and glucose—the brain uptakes about 20% of whole resting body glucose mass, although it constitutes just about 2% of the body mass [24].

The main pathway of energy production in the brain is ATP synthesis from glucose in glycolysis, tricarboxylic acid (TCA) cycle, and oxidative phosphorylation. Cytoplasmic glycolysis splits a molecule of glucose into two pyruvate molecules that are converted to acetyl CoA (acetyl coenzyme A) in an aerobic environment (Figure 1). Acetyl CoA enters the mitochondria matrix where it undergoes the TCA cycle, resulting in the generation of ATP, NADH (nicotinamide adenine dinucleotide), and FADH_2_ (dihydroflavine adenine dinucleotide), donating electrons and protons to the mitochondrial ETC. A series of reactions in ETC with the final accepting of electrons by oxygen (oxidative phosphorylation) produces a lot of energy which is converted to ATP. Phosphocreatine (creatine phosphate) is an important element in the brain energy metabolism, as it is a “ready-to-use” reserve of high-energy phosphates to recycle ATP (reviewed in [25]). Glycolytic and TCA intermediates participate in the biosynthesis of carbohydrates, neurotransmitters, neuromodulators, and amino acids [7].

Brain metabolism is differently regulated in neurons and astrocytes [26]. Neurons favor oxidative metabolism, whereas astrocytes are glycolytic cells. It was suggested that glutamate released by neurons into synapses was taken by astrocytes, in which glutamate stimulated glycolysis and the production of lactate, which can be oxidized in neurons [27].

Brain energy deficit was suggested by the ^31^P-NMR study, in which a lower phosphocreatine to creatine ratio, and enhanced concentrations of ADP were observed (reviewed in [28]). Several other effects associated with energy deficit in migraine, including decreased levels of ATP and ATP hydrolysis-driven energy in the occipital cortex of migraineurs [29,30]. Importantly, the degree of energy deficit was positively correlated with the severity of the attack [31].

Electrophysiological, neurochemical, genetic, and pharmacological studies support the picture of a hypersensitive brain in migraineurs [32,33]. This hypersensitivity results in enhanced cortical response to sensory and attentional demands, requiring enhanced energy utilization by the brain [34,35]. On the other hand, migraine associates with mitochondrial diseases, but this association is independent of the clinical picture attributed to exact mitochondrial disorders, suggesting that this association is underlined by the intrinsic features of mitochondria changed in a particular disease [5,36]. Such changes are likely linked with decreased energy production.

Migraine associates with several other effects and phenomena, resulting in lower glucose levels, or impeding its uptake and utilization, indicating a brain energy deficit reported in migraineurs, such as impaired glucose transport across the blood-brain barrier (BBB), hypoxia, and hypoglycemia [8,37,38,39].

Several pathways from impairment in the energy production to migraine are considered, including cortical spreading depression (CSD) connection, direct irritation of meningeal nociceptors by lactate, and an energy impairment-induced hyperexcitability of the trigeminovascular system (reviewed in [14]). However, the exact role of these pathways in migraine pathogenesis is not clear. On the other hand, energy deficit driven by mitochondrial impairment is associated with oxidative stress with ROS overproduction. These ROS and other oxidants can be sensed by TRPA1 (transient receptor potential ankyrin-1) ion channels present in nociceptive nerve endings [40]. Activated TRPA1 can induce pain signaling and neurogenic inflammation, symptoms featuring in migraine attacks (reviewed in [41]). Therefore, impaired mitochondria, decreased energy production, and oxidative stress can be considered as mutually dependent components of a functional complex that may be essential in migraine pathogenesis. Consequently, they can be targeted in migraine preventive and therapeutic strategies.

## 3. Mitochondria in Migraine Pathogenesis

As mitochondria are the main objects to produce energy in the brain, any mitochondrial impairment is likely associated with energy deficit, which, in turn, can be a migraine trigger.

Riboflavin (vitamin B2) links migraine with mitochondria, as it is important for vital mitochondrial processes, and is effective in migraine prophylaxis in experimental studies and clinical trials (reviewed in [42,43]). Several mutations in mtDNA associate with the occurrence of migraine or migraine-like syndromes (reviewed in [44]). However, it should be underlined that complex mitochondrial diseases cannot be directly and exclusively related to mtDNA, as they are associated with defects in mitochondrial proteins encoded by nuclear DNA. This statement has a more general significance and should be related to all associations between migraine and pathology, as the autonomy of the mitochondrial genome is very limited, and mitochondrial homeostasis is determined almost entirely by the expression of its nuclear counterpart. Fila et al. presented arguments for the possible role of the mtDNA epigenetic profile in migraine pathogenesis, but this issue should be explored in further studies [6].

The role of mitochondrial dysfunctions in migraine pathogenesis is underlined by migraine-related mechanisms triggered by several effects, leading to energy deficit in neurons and astrocytes, such as impaired oxidative phosphorylation, associated with an increased production of reactive oxygen and nitrogen species, as well as an increased calcium influx (reviewed in [45]). These effects are marked by the low activity of mitochondrial superoxide dismutase (SOD2), activation of cytochrome-c oxidase and nitric oxide, high levels of lactate and pyruvate, and low ratios of phosphocreatine-inorganic phosphate and *N*-acetylaspartate-choline (Figure 2).

The mitochondria-migraine association is supported by some nutrients whose intake influences mitochondrial homeostasis, and has a beneficial effect in migraine. Some of them will be discussed further in this review.

## 4. Oxidative Stress in Migraine

Mitochondrial ETC produces ROS even in its normal functioning, and this production may increase when ETC is impaired, which may lead to the mitochondrial vicious cycle [46].

The activity of the brain requires energy, which is provided mainly by oxidative metabolism [47]. Oxygen is also needed for catecholamine neurotransmitter synthesis, and degradation by tyrosine hydroxylase, dopamine-β-hydroxylase, tryptophan hydroxylase, and monoamine oxidase [48]. Synthesis of serotonin and acetylcholine is strongly sensitive to hypoxia [49]. When impaired, intense oxygen flow may cause oxidative stress, associated with an increased production of reactive oxygen and nitrogen species that may damage biological macromolecules, including proteins, nucleic acids, and lipids.

The brain, as other organs and tissues, has the antioxidant defense system to cope with increased oxidative stress. This system includes antioxidant enzymes, DNA repair proteins, and small molecular weight antioxidants.

Gross et al. observed that the majority of 32 higher frequency episodic migraineurs had lower levels of alpha-lipoic acid (LA, thioctic acid), and almost half of patients showed abnormal levels of lipid peroxides in their blood serum [50]. Total plasma antioxidant capacity was lower in one-third of the patients. This research related the results obtained in migraineurs to the local normative values without a comparison with a control group. Moreover, the population of the patients was heterogenous in sex, age, and other social parameters, as well as in migraine-related characteristics, such as aura occurrence or disease prophylaxis.

## 5. Usefulness of Nutrients Targeting Energy Production in Migraine Prevention

Some kinds of diet that are recommended to follow or avoid in migraine prevention, and treatments seem to support the usefulness of dietary interventions targeting one or more of elements of the mitochondria dysfunction/decreased energy production/oxidative stress triad in migraine prevention. For instance, the ketogenic diet, a low-carbohydrate and high-fat diet was reported to be beneficial in migraines and stimulates mitochondrial metabolism [51,52,53,54,55,56]. However, each specific diet, including the ketogenic diet, has many variants, with components different from each other that may interact, producing different outcomes than expected. Therefore, it is somehow risky to recommend a complex diet that would be applied in a particular disorder, and a more reasonable approach may be to adjust a general “healthy diet” to a specific disease by the elimination of products containing nutrients that are known to support the disease, or by supplementation with products containing nutrients which may have a beneficial effect for individuals affected by or at risk of the disease.

In this section, we present some nutrients that may be recommended to supplement a general “healthy” diet to prevent/treat migraine and its relationship to mitochondria/oxidative stress/energy production. In addition, we give a short note on caffeine and alcohol consumption in relation to migraine.

### 5.1. Riboflavin

In the nervous system, riboflavin is crucial for the synthesis of myelin, and its deficiency may contribute to the disruption of myelin lamellae (reviewed in [57]). As mentioned, riboflavin linked migraine with mitochondria, as it was reported to play an important role in migraine pathogenesis and mitochondria maintenance. Recently, Yamanaka et al. published a narrative review on the efficacy of riboflavin in migraine prophylaxis, concluding that this agent has the potential to prevent migraine attack, but experimental and clinical evidence is too weak to draw a definitive conclusion [43]. However, there is no evidence that riboflavin deficiency is associated with migraine headaches, which is important in light of a significant world-wide proportion of individuals with genetically determined impairment in riboflavin absorption and utilization [58].

Several clinical trials confirmed the efficacy of riboflavin for migraine prevention, and it is currently classified as a Level B medication for migraine according to the American Academy of Neurology evidence-based rating, with evidence supporting its efficacy (reviewed in [59]).

The application of riboflavin in both adults and adolescent migraine prophylaxis was inspired by a role of a deficit of mitochondrial energy metabolism in migraine pathogenesis [60,61]. The potential mechanism of riboflavin protective effects is explained by its involvement in the regulation of the mitochondria-energy production-oxidative stress pathway, but the details of this involvement are not known. DiLorenzo et al. showed that riboflavin was more effective in patients with migraine with non-H mitochondrial DNA haplotypes [62]. The authors explained obtained results by the association of haplogroup H with increased activity in complex I, a major target for riboflavin. These results may also contribute to ethnic implications of migraine, as the haplogroup H is mainly found in the European population. The other mechanism underlying protective effects of riboflavin in migraine is reduction of neuroinflammation, which lies at the heart of the neurogenic basis of migraine [63,64] (Figure 3).

Riboflavin affects processes of energy generation in mitochondria, including the TCA cycle, oxidative phosphorylation, and metabolism of amino acids, fatty acids, and nucleotides [65,66]. These processes depend on flavoenzymes, including oxidases, reductases, and dehydrogenases. Flavoenzymes functionally depend on their redox cofactors flavin adenine dinucleotide (FAD) or flavin mononucleotide (FMN), for which riboflavin is their precursor [67]. Therefore, riboflavin is an essential element in antioxidant defense exerted by flavoenzymes protecting the cells from antioxidant stress and apoptosis.

Riboflavin may also ameliorate the mitochondria/energy production/oxidative stress axis by preventing mitochondrial ROS production and release of mtDNA, which activates the NLRP3 (NLR family pyrin domain containing 3) and non-canonical inflammasomes [43]. Also, riboflavin inhibits the activity of caspase-1, NLRC4 (NLR family CARD domain containing 4), and AIM2 (absent in melanoma), a protein important for several cellular activities, including cell proliferation [68]. Therefore, riboflavin may play an important role in migraine prevention through its involvement in antioxidant and anti-inflammatory reactions due to mitochondrial dysfunctions.

In short, riboflavin is an important compound for mitochondrial homeostasis, energy production, and protection against oxidative stress in the brain, and its preventive potential in migraine was confirmed in clinical trials.

### 5.2. Thiamine

Thiamine (vitamin B1) deficiency was reported in several human pathologies, including the most remarkable syndrome of its deficiency, beriberi (reviewed in [69]). Therapeutic potential of thiamine in migraine has a long-standing history, as its successful use in headache treatment was reported in 1949 [70,71].

Mammalian cells obtain thiamine from the environment and change it to thiamine pyrophosphate (TPP) in the cytoplasm [72]. Most of TPP is transported into mitochondria with the involvement of the mitochondrial thiamine pyrophosphate transporter (MTPPT).

Thiamine may play a role in the propagation of nerve impulses and myelin sheath maintenance, which are essential in migraine pathogenesis, but the mechanisms underlying this role are not known [69,73].

Thiamine was reported to improve the symptoms of chronic cluster headache in a case study, confirming its role in pain modulation [74]. In another case study, low blood levels of thiamine in two female migraine patients were observed [75]. Also, clinical signs pertinent with a diagnosis of Wernicke’s encephalopathy (WE) were observed in these patients. Intravenous thiamine supplementation ameliorated headaches and WE. As WE in its early stage simulates migraine symptoms, the authors speculated that thiamine deficiency due to nausea, vomiting, and anorexia in migraine might further worsen migraine headaches in a cyclical pattern, and thiamine supplementation might break this cycle, presenting a promise in migraine therapy.

In a case-control study, Faraji et al. showed that dietary intake of thiamine in migraine patients was lower than in controls [76]. Moreover, the consumption of thiamine in the patients negatively correlated with the severity of migraine attacks. However, these associations were not significant after the adjustment on energy intake. Therefore, that study confirmed an important role of energy balance in migraine, and a potential involvement of thiamine in its pathogenesis through its role in brain energy metabolism. Parker et al. isolated intact mitochondria from the brain and liver of rats on a low-thiamine diet [77]. They observed abnormalities in mitochondrial ATP synthesis associated with depression of the activity of the brain and liver pyruvate dehydrogenase complex, and alpha-ketoglutarate dehydrogenase complex in the thiamine-deficient group. Therefore, observed mitochondrial abnormalities resulting from thiamine deficiency might be secondary to the depression of thiamine-mediated enzyme activity.

In summary, thiamine is an essential human nutrient, whose metabolic product is largely transported to mitochondria, where it may be implicated in mitochondrial functions. Thiamine may play a role in pain modulation, and its deficiency may be linked with migraine occurrence.

### 5.3. Coenzyme Q10

Coenzyme Q 10 (CoQ10, ubiquinone) is involved in many cellular redox reactions important for bioenergetics and antioxidant defense [78]. The major cellular function of CoQ10 is electron transport in ETC. CoQ transports electrons between either complex I or II to complex III by accepting electrons from NADH or succinate, respectively [79]. CoQ10 is involved in ROS production. The accumulation of reduced CoQ10 generates superoxide through the retrograde electron transport, guiding electrons throughout complex I to NAD+ [80]. The best-known antioxidant action of CoQ10 is inhibition of lipid peroxidation and regeneration of the active form of vitamin E [81]. CoQ10 also exerts a protective effect against oxidative DNA damage [82]. Therefore, CoQ10 links mitochondrial functioning with energy production and oxidative stress. It was observed that CoQ10 reduced tacrolimus-induced oxidative stress, and protected mitochondria in pancreatic beta cells [83].

Several controlled randomized trials suggest that CoQ10 is a potential safe complementary agent, with some evidence of efficacy in the management of migraine, as was summarized by Paraohan et al. [84]. The most pronounced impact of CoQ10 is the reduction of frequency of migraine attacks. In a recent meta-analysis, Sazali et al. included six randomized control trials comparing CoQ10 with a placebo or used as an adjunct treatment [85]. There was no significant reduction in the severity of migraine headache after CoQ10 supplementation, but there was a reduction in the duration of headache attacks, and the frequency of migraine headaches.

It was observed that an 8-week combined supplementation with CoQ10 and L-carnitine reduced the severity of migraine attacks, their duration, frequency, headache diary result (HDR), as well as serum levels of lactate in migraineurs as compared with placebo-receiving controls [86]. A significant effect of 8-week nano-curcumin and CoQ10 supplementation in migraineurs on frequency, severity, duration of migraine attacks, and HDR as compared to groups of patients receiving nano curcumin or CoQ10 singly or the placebo group was observed [87]. Also, the combined group had better scores in migraine-specific questionnaires at the end of the study compared to other groups. No side effects were reported by the patients enrolled in the study.

It was shown that 61.3% of 32 patients treated with 150 mg of CoQ10 per day had a greater than 50% reduction in the number of days with migraine headache [88]. The mean number of days with headache during the baseline period was 7.34, and decreased to 2.95 after 3 months of treatment. A reduction in migraine attack frequency was also observed, with no side effects. A similar conclusion was drawn in another study analyzing five studies with CoQ10 supplementation in migraineurs [89].

In summary, several randomized clinical trials suggest that the supplementation with CoQ10 may be beneficial in migraineurs in respect to reduction headache severity, frequency, and duration of headache attacks, with no side effects. Considering the well-established role of CoQ10 in mitochondrial energy production and oxidative stress prevention, it is somewhat surprising that there is a lack of studies on the potential mechanisms of the beneficial effects of CoQ10 in migraine.

### 5.4. Magnesium

Magnesium is the fourth most abundant cation in the human organism and is involved in over 600 enzymatic reactions occurring in the human organism, including those in energy production [90]. Magnesium is needed for the synthesis of compounds with energy-rich bonds to allow energy storage as high-energy phosphates [91].

Magnesium inhibits calcium influx in neurons by blocking *N*-methyl-d-aspartate (NMDA) receptors that play a role in the initiation and maintenance of central sensitization after nociceptive stimulation (reviewed in [92]). NMDA receptors may facilitate cortical spreading depression and be a target in anti-migraine treatment [93,94]. Blocking calcium channels in neurons may contribute to the inhibition of intracellular pro-inflammatory signaling, related to migraine pathogenesis [95]. Several other mechanisms have been indicated to associate magnesium deficit with migraine, including magnesium link with CSD, imbalance release of neurotransmitters, platelet activity, and vasoconstriction (reviewed in [96]). Moreover, magnesium was shown to decrease CGRP (calcitonin gene-related peptide), which is targeted by recent anti-migraine drugs with the hope for an outbreak in migraine therapy [97,98,99,100].

There is vast literature on the efficacy of magnesium supplementation in migraine, which cannot be detailly reviewed here. Only some recent studies are presented.

Migraine patients were shown to have lower serum levels of magnesium as compared with the non-migraineurs control, and migraine patients with severe headaches had lower magnesium levels than their counterparts with mild to moderate headaches [101]. It should be underlined that all individuals enrolled in that study had serum magnesium concentrations within physiological range.

Sodium valproate is commonly accepted in migraine prophylaxis, with reports on the reduction in the number of migraine attacks or days with migraine by about 50% in about half of migraine patients [102]. Khani et al. evaluated the efficacy of the combined magnesium-sodium valproate therapy and compared it with either magnesium or sodium valproate singly in migraine prophylaxis [103]. They observed that supplementation with magnesium in sodium valproate-treated patients increased the anti-migraine action of sodium valproate, and reduced its dose needed for efficient anti-migraine treatment.

The measurement of magnesium is challenging for several reasons (reviewed in [104]). Moreover, there are concerns on the cases of the actual relationship between a disease and the concentration of an agent in non-target tissue/organs. However, the magnesium load test showed a larger magnesium retention in migraine patients than non-migraine controls, suggesting a systemic deficiency in magnesium in migraineurs [105]. On the other hand, lowered magnesium levels in migraineurs were observed in cerebrospinal fluid, and the ictal and interictal regions within the brain and salivary glands [106,107,108]. Although not easy to measure, magnesium deficiency is considered as an independent risk factor for migraine (reviewed in [96]).

Barbiroli et al. studied the occipital lobes of patients with mitochondrial cytopathies with ^31^P-NMR [109]. All patients showed impaired mitochondrial respiration, low phosphocreatine (PCr) and free Mg^2+^ concentrations, and high concentrations of ADP and inorganic phosphate (P_i_). Treatment with CoQ10 improved the efficacy of the respiratory chain, as shown by an increased PCr, and a decreased P_i_ and ADP, and increased the availability of free energy and cytosolic free Mg^2+^. The authors concluded that low brain free Mg^2+^ resulted from failure of the respiratory chain and, consequently, lower available free energy. These results were confirmed in the next studies of the same team that used ^31^P-NMR, and showed that cytosolic free magnesium ions, as well as the free energy released by ATP hydrolysis in the occipital lobes, were significantly reduced in patients with migraine in attack-free periods [29]. These parameters displayed a trend to associate with the severity of migraine syndromes, with the lowest concentration of magnesium in migraine stroke, and the highest in migraine without aura.

It was shown that the supplementation with a fixed combination of magnesium, riboflavin, feverfew, andrographis paniculata, and CoQ10 were an effective and well-tolerated preventive approach against episodic migraine [110].

In short, magnesium is essential for brain energy homeostasis, and its deficiency is an evidenced risk factor for migraine.

### 5.5. Melatonin

Melatonin (*N*-acetyl-5-methoxytryptamine), an indole produced mainly by the pineal gland and gastrointestinal tract, is one of the agents to have the greatest number of effects, mostly positive, on cell and human biology (reviewed in [111]). Its best-known physiological function is to regulate the central circadian clock, which, in humans, is in the suprachiasmatic nuclei of the hypothalamus (reviewed in [112]). Melatonin was reported to have many mitochondria-related effects, including those directly associated with energy metabolism (reviewed in [113]). It is also an effective ROS and RNS (reactive nitrogen species) scavenger (reviewed in [114]).

The role of melatonin in sleep regulation directly implies its involvement in migraine, as sleep disturbances are known to associate with migraine (reviewed in [115]). Pellegrino et al. showed in a meta-analysis that sleep disorders were the second most common trigger in primary headaches [116]. In these regards, melatonin treatment represents a hopeful therapeutic strategy for migraine comorbid with sleep disorders. Also, some mitochondrial diseases are linked with sleep disorders. However, no evidence of a direct link between the mitochondrial functions of melatonin and sleep-disorders-associated migraine was presented [117,118].

Melatonin had mostly a beneficial effect in clinical trials, applied singly or in combination with amitriptyline, an antidepressant and antimigraine drug [119,120,121,122]. Also, observational studies reported the efficacy of melatonin in migraine prevention [123,124,125]. Only a small fraction of individuals enrolled in these studies reported adverse effects, which were usually mild and tolerable [126]. However, more studies on melatonin safety are warranted, as some its effects could escape researchers’ attention, as they could be masked by melatonin formulation or inadequate research design.

In summary, melatonin reduced headache frequency, duration, and intensity in several clinical trials and observational studies with little, if any, adverse effects. Although no studies were directly related to mitochondrial functions and/or energy production, one can speculate that they may be involved, at least in part, in the beneficial effects of melatonin in migraine, given its established role in these processes.

### 5.6. Niacin

Niacin (vitamin B3, vitamin PP) refers to two compounds, nicotinic acid and nicotinamide, that are nutritional precursors of nicotinamide adenine dinucleotide (NAD) and nicotinamide adenine dinucleotide phosphate (NADP). They are cofactors or cosubstrates for many enzymes that are involved in cellular homeostasis, and the pathogenesis of several human diseases (reviewed in [127]). In particular, the NAD/NADH ratio is important for essential pathways of cellular metabolism, as it regulates the activity of enzymes central to glycolysis, TCA cycle, and fatty acid oxidation and synthesis. Deficiency of dietary niacin may cause reduced oxidative phosphorylation, and impaired mitochondrial respiration [128]. There are more mechanisms of direct or indirect influence of niacin on mitochondrial functions, including mitochondrial sirtuins, PARPs (poly (ADP-ribose)), polymerases, and mitophagy (reviewed in [127]).

In a case study, it was observed that administration of sustained-release niacin significantly improved the state of a patient with migraine headache attacks [129]. The authors hypothesized that niacin acted as a negative feedback regulator on the kynurenine pathway of tryptophan metabolism, directing it to the serotonin pathway, of which low levels are reported in migraineurs [130]. On the other hand, riboflavin, which has beneficial properties in migraine, has just been presented, and may help to convert tryptophan into niacin [131]. Gedye hypothesized that a combined administration of low doses of tryptophan, niacin, calcium, caffeine, and acetylsalicylic acid (ASA) shortly after migraine attack might be beneficial [132]. This hypothesis was supported by the observation that such a combined treatment resulted in beneficial effects in 9 out of 12 migraine patients.

Prousky and Seely critically reviewed papers on the application of niacin in migraine and tension-type headaches until 2004 [133]. They did not find any randomized or controlled trials of niacin in migraine. They concluded that most, if not all, articles they analyzed contained serious flaws, such as lack of proper control, lack of basic information about patients and treatment, no methods of evaluating treatment efficiency, and others. The authors concluded that at that time (2005), some data suggested that niacin might have a therapeutic effect on migraine headaches, but randomized trials were required to evaluate the clinical value of niacin treatment. The mechanism underlying the potential beneficiary effects of niacin in migraine was uncertain and could be only hypothesized. Prousky and Seely mentioned the potential of niacin as a peripheral vasodilator, the properties of which could be extended on central vasodilation, which may play a role in migraine pathogenesis [134]. The other reason of the effectiveness of niacin in migraine prevention was its improvement of brain energy deficit through augmentation of the complex I of ETC.

Despite some promising information on the potential impact of niacin in migraine treatment, and calls for randomized clinical trials, no further studies on this subject have been performed.

In brief, niacin is indirectly involved in the glycolysis pathway of energy production in the brain, and its deficiency impairs mitochondrial functions. Some studies suggest a promising potential of niacin in migraine prophylaxis, but both experimental research and controlled clinical trials are needed to provide more rationales for its use in migraine treatment.

### 5.7. Carnitine

Carnitine (l-3-hydroxy-4-*N*-trimethlaminobutyric acid) is an essential nutrient, which is only in a small part produced endogenously, whereas the majority is obtained from diet, mainly animal products (reviewed in [135]). It is involved in the metabolism of fatty acids facilitating their transfer to the mitochondrial matrix where they are β-oxidized to acetyl-CoA entering the TCA cycle and contributing to ATP production (Figure 1). Therefore, carnitine is involved in cellular energy production in mitochondria.

Few studies addressed the potential of carnitine in migraine. It was shown, in a single-blinded clinical trial, that migraineurs whose diet was supplemented by L-carnitine, had a decreased ratio of migraine attacks, although the severity of the symptoms was unchanged [136]. In a two-case study, Kabbouche et al. showed that adolescent girls diagnosed with migraine had a low level of creatinine, and several side symptoms resulting from their prophylactic treatment. Carnitine supplementation resulted in reduction in headache frequency, and amelioration of other symptoms. A muscle biopsy from one patient revealed partial deficiency of carnitine palmityltransferase I, an enzyme involved in carnitine metabolism.

Hagen et al. did not observe any difference in headache outcomes between patients treated with acetyl-carnitine, and controls in a randomized triple-blinded placebo-controlled cross-over study [137]. Beneficial effects of L-carnitine and coenzyme Q-10 on the severity, duration, and frequency of headache attacks, and the headache diary results were observed in migraine patients in a randomized, placebo-controlled, double-blind trial [86]. In addition, a reduction in lactate serum levels was observed in treated patients, indicating an improvement of mitochondrial oxidative metabolism, but recent data suggest that lactate may induce a hormetic protective effect against oxidative stress [138]. In a recent case study, Charleston et al. observed a carnitine deficiency in a patient with chronic migraine-like headaches [139]. Carnitine supplementation significantly improved patient headaches, and the authors concluded that carnitine deficiency might be in the differential in a refractory migraine.

In short, carnitine plays a role in fatty acid metabolism and oxidation in mitochondria, contributing to energy production. Some studies provide evidence on carnitine deficiency in headache-suffering individuals, and on ameliorating such aliments after carnitine supplementation.

### 5.8. Lipoic Acid

Lipoic acid has antioxidant properties, regenerates other antioxidants, and improves mitochondrial functions, including the activity of mitochondrial superoxide dismutase [140,141].

Ali et al. used LA singly or in combination with topiramate, an oral drug with evidenced efficacy in migraine prevention, to treat 40 secondary school girls suffering from migraine [142]. The authors observed a reduction in the mean monthly migraine attack frequency in all treatment regimes, but the most pronounced changes were for combined therapy, although the dose of topiramate was just half of the commonly used dose at that time. Moreover, topiramate was better tolerated when administrated with LA than in monotherapy.

A reduction in migraine attacks, and days of treatment were observed in patients with insulin resistance treated with LA in addition to their on-going treatment [143]. Glucose and insulin levels, quantitative insulin sensitivity check index, and the Stumvoll index did not change after 6 months of LA treatment as compared with baseline.

It was suggested that LA might be beneficial in migraine prophylaxis, based on a randomized double-blind placebo-controlled trial in 44 migraine patients [144]. However, these studies, besides being conducted on relatively small cohort, presented some results of a border-line significance, but reduction in the attack frequency and headache severity were significant in contrast to placebo group.

As previously mentioned, the TRPA1 channels are important for CSD, which may be critical for migraine pain [145]. An LA-based TRPA1 antagonist, ADM_90, was shown to have the potential to rapidly cross BBB in the neocortex by its formulation with niosomes, nanovesicles that can be functionalized with specific ligands. A 3-month supplementation with ALA in patients with episodic migraines resulted in an improvement of inflammatory and oxidative conditions, as well as mood disorders [146].

Gross et al. observed in a cross-sectional study that about 90% of 32 higher frequency migraineurs patients had abnormally low levels of LA [50]. The authors also detected changes in some markers of mitochondrial dysfunction and oxidative stress. An abnormally low level of lactate was observed in the serum of the majority of the patients. This study suggests a potential of LA as a migraine marker, and further confirms the role of mitochondrial metabolism in migraine pathogenesis.

In summary, LA is a powerful antioxidant with the potential to ameliorate oxidative stress and inflammation in migraine, and its deficiency may be considered as a migraine marker. Randomized trials on its usefulness in migraine prevention should be continued, as such initial studies brought some promising results.

### 5.9. Pyridoxine, Folate, and Cobalamin

Migraine was associated with vascular comorbidities, including stroke [147]. Lowering the levels of homocysteine (Hcy) was reported to exert protective effects against detrimental vascular events that were often characterized by hyperhomocysteinemia (reviewed in [148,149]). Several studies showed a correlation between serum Hcy levels and the frequency and characteristics of migraine attacks (reviewed in [150]). Moreover, the mechanism of the involvement of Hcy in vascular disorders, including stroke, may be related to folate cycles and remethylation of Hcy to methionine in the central nervous system (reviewed in [151]). We previously suggested that these effects might play a role in migraine pathogenesis [152]. Elevated levels of Hcy were associated with disturbances in the functioning of mitochondria, and energy production in the central nervous system (reviewed in [153]). Sadeghi et al. did not find any correlation between serum Hcy levels and characteristics of migraine, including severity, duration, frequency, and HDR, in a cross-sectional study with 120 patients [154]. However, such a correlation between serum Hcy levels and HDR was observed in male patients after adjustment for age, BMI, and family history of migraine. Dietary compounds, pyridoxine (vitamin B6), folic acid, and cobalamin (vitamin B12) were shown to lower Hcy levels in many pathologies [151,155,156,157]. Moreover, these compounds are also involved in mitochondrial homeostasis, energy production, and antioxidant defense in the central nervous system [158,159,160]. Several studies suggest a beneficiary effect of pyridoxine, folate, and cobalamin supplementation in migraineurs. Sadeghi et al. observed that pyridoxine intake (80 mg/day) in 66 patients with migraine with aura resulted in beneficiary effects on headache severity, HDR, and migraine attack duration, but not their frequency [161]. A significant decrease was seen in migraine severity, frequency, and duration of migraine attacks, as well as in HDR in 95 patients with migraine with aura with a diet supplementation with pyridoxine (80 mg/day) and folic acid (5 mg/day) for 3 months, as compared with placebo or folic acid only [162]. Lea et al. administered, for 6 months, 2 mg of folic acid, 25 mg pyridoxine, and 400 µg cobalamin daily to 52 patients with migraine with aura [163]. In that study, a moderate effect of the genotypes of the c.677C>T polymorphism in the MTHFR (methylenetetrahydrofolate reductase) gene was also observed. This polymorphism was reported to associate with migraine susceptibility in numerous studies (reviewed in [164]). The same group observed, in a subsequent study, that lowering the dose of folic acid to 1 mg/day resulted in a less effective reduction of migraine syndromes after the combined folic acid/pyridoxine/cobalamin supplementation [165]. A case-control study on 70 migraine patients and 70 individuals without headache showed that individuals with a lower status of cobalamin and methylmalonic acid had higher odds for migraine [166].

In summary, many studies suggest that supplementation with folic acid, pyridoxine, and cobalamin may exert a beneficial effect in migraine, first in adults with aura. Therefore, further studies with other age groups are required. Moreover, such dietary intervention should be studied with genotyping of the c.677C>T polymorphism in the MTHFR gene, as many studies suggest its involvement in migraine pathogenesis.

### 5.10. Caffeine and Alcohol

Caffeine may stimulate ETC by restoring the activity of complex IV, as was shown in septic rats by Verma et al. [167]. Tea and coffee contain thiaminases, enzymes degrading thiamine [69]. As stated above, thiamine may have a protective potential in migraine, and caffeine can be considered as a migraine trigger. On the other hand, migraine may be triggered by caffeine withdrawal (reviewed in [2]). At present, these two apparently opposite effects cannot be directly related to the brain energy balance. Therefore, caffeine, although it may be potentially beneficial for mitochondrial functions, cannot be recommended to ameliorate migraine symptoms, unless it is not withdrawn after a long use.

Although alcohol is not an essential nutrient, and even impairs the absorption of nutrients by damaging the mucosa of gastrointestinal cells and weakening the transport of many nutrients into the blood, there are many reports showing its role in migraine pathogenesis, and it is an energy substrate also for the brain (reviewed in [168]). It is sometimes called a “nonnutrive nutrient”.

Alcoholic beverages, especially red wine, are recognized by migraine patients as a potential migraine trigger [169]. Alcohol may induce or increase oxidative stress through the generation of ROS and RNS [170]. Reddy et al. observed that the administration of alcohol to rats increased the mitochondrial lipid peroxidation, protein oxidation, and nitric oxide levels in their brain cortex [171]. These authors also observed decreased SOD2 levels at both mRNA and protein expression levels. The authors interpreted their results as caused by decreased activities of Na^+^/K^+^-ATPase and complexes I, III, and IV of the ETC, as well as decreased content of cardiolipin in the brain of intoxicated animals. Surely, these results do not directly evidence the involvement of mitochondrial dysfunction in migraine pathogenesis, but they do support the link between alcohol, migraine, and mitochondria.

In summary, both caffeine and alcohol may have different effects on migraineurs, depending on their “history of the use”. Therefore, the intake of both substances should be adjusted to individual needs, and any general recommendation type use/not use is not currently justified.

## 6. Conclusions and Perspectives

Studies of the brain of migraineurs with ^31^P-NMR revealed deficient energy production in the structures linked with the disease pathogenesis, including central pain processing areas [172]. These studies showed that migraine is a disease with decreased energy production, and increased demand. As the mitochondrial oxidative pathway of glucose processing is the main route of energy generation, it is justified to relate mitochondrial functions and oxidative processes, including oxidative stress to migraine pathogenesis. This, in turn, raises the question of which nutrients, besides carbohydrates, may be specifically important in the brain energy balance by assisting mitochondrial functions and oxidative stability. As many dietary compounds are reported by migraineurs as migraine triggers, or have a beneficial effect for migraine headaches, it is reasonable to check whether these nutrients that are important for the brain energy balance by their role in migraine functions and oxidative stress modulation, may be efficient in migraine prevention and treatment.

The diet is a “hot topic” in migraine, and surely will be for a long time [15]. However, some anti-migraine drugs targeting the CGRP signaling pathway, including monoclonal antibodies against the CGRP receptor, or ligand small molecule CGRP receptor antagonists, such as ditans and gepants, are considered as drugs that may make a breakthrough in migraine treatment [173,174,175,176]. Therefore, it is possible these drugs will resolve all the problems associated with migraine. However, it is too early to draw a definitive conclusion on the efficacy of these drugs. Secondly, there are some concerns on the long use of them, as the consequences of targeting CGRP in organs other than the brain are unknown [177]. These drugs are expensive, which is an important factor to consider in their long-time use. Altogether, these drugs should be considered in refractory migraine rather than as a first-line treatment [178]. But first, it is always better to prevent than to cure. Therefore, diet modifications with nutrients that may be beneficial for migraineurs are still required. In this paper, we presented some nutrients that are known to have a beneficial potential in migraine. They are also involved in energy production in the brain, and may influence mitochondrial function, and modulate oxidative stress. However, the question is about the basic diet which might be modified by the nutrients we presented. We are skeptical of various specific diets, such as the “migraine diet”, “mito diet”, or “epigenetic diet” [152]. We consider that a basic diet should be adjusted to individual constitution and needs, and then fine-tuned with specific nutrient(s) to a specific condition, such as being overweight, having an organic disease, and others. It should be underlined that this basic diet is individual and should not be recommended for all.

Among the nutrients we considered, riboflavin seems to be closely connected with migraine and mitochondrial energy production. Its beneficial potential in migraine has been confirmed in controlled clinical trials. Also, CoQ10 is worth further studies due to the basic role it plays in the mechanism of energy generation in the brain. All other compounds require further research; specifically, controlled placebo-based clinical trials to confirm their efficacy and safety in migraine prevention and/or treatment.

In conclusion, several nutrients can be considered to supplement the diet to prevent and/or ameliorate migraine through the improvement in brain energy production by the amendment of mitochondrial functions, and/or the decrease of oxidative stress (Figure 4). These are riboflavin, thiamine, magnesium ions, niacin, carnitine, coenzyme Q10, melatonin, lipoic acid, pyridoxine, folate, and cobalamin. They can supplement a normal, healthy diet, which should be adjusted to individual needs determined mainly by the physiological constitution of the individual. The intake of caffeine and alcohol should be tuned to the history of their use—withdrawal of these agents in a regular user may become a migraine trigger.

## Figures and Tables

**Figure 1 nutrients-13-04433-f001:**
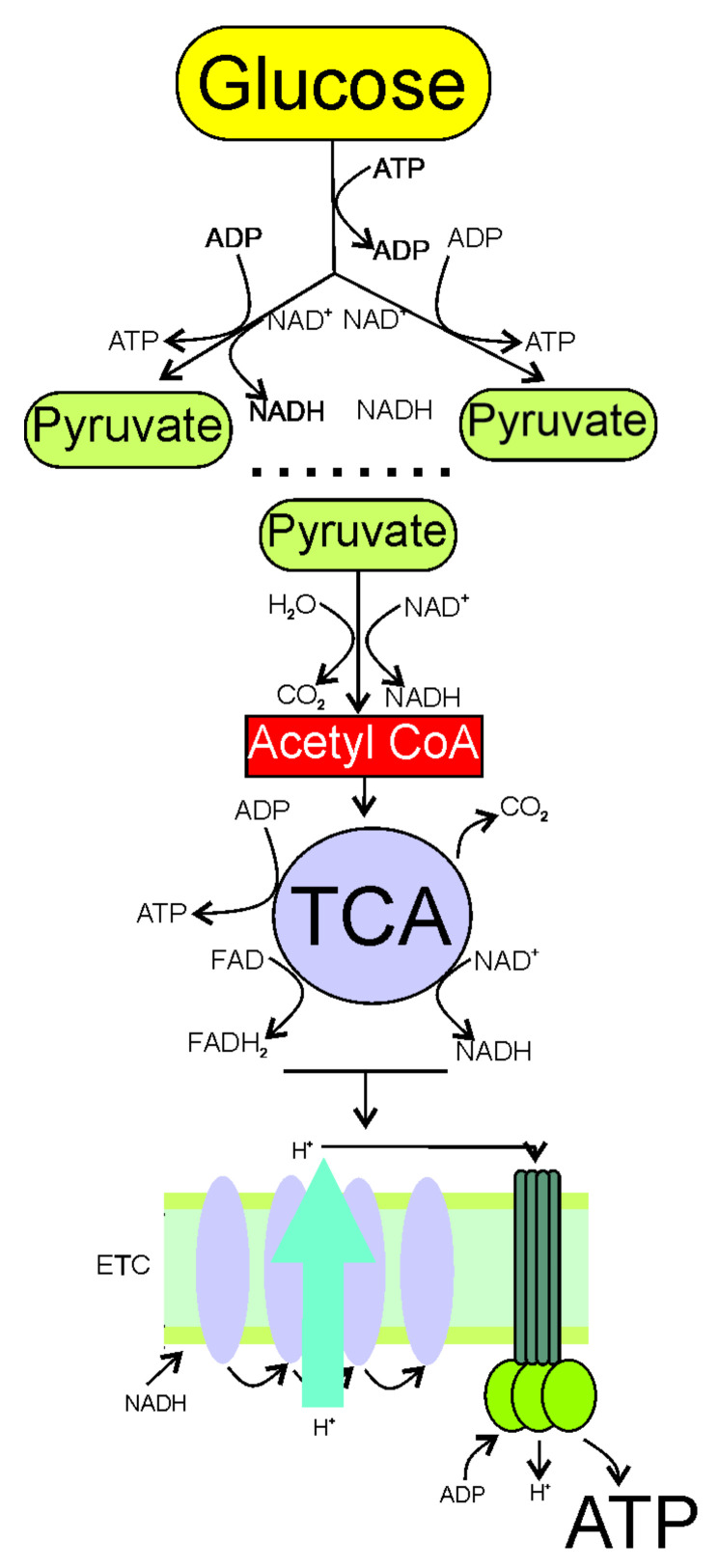
Synthesis of ATP from glucose, the main process of energy production in the brain, consists of three essential stages: glycolysis; the citric acid cycle (TCA); and oxidative phosphorylation. Glycolysis occurs in the cytoplasm, and uses ATP to split glucose into two pyruvate molecules. This process gives a net gain of two ATP molecules. In an aerobic environment, each pyruvate becomes acetyl CoA (acetyl coenzyme A), and goes to the mitochondria matrix to perform the TCA cycle, resulting in the production of CO_2_, NADH (nicotinamide adenine dinucleotide), FADH_2_ (dihydroflavine adenine dinucleotide), and ATP. The generation of coenzymes NADH and FADH_2_ is critical, as they donate electrons and hydrogen to ETC in the inner mitochondrial membrane within the cristate to produce more ATP—one molecule of glucose gives 36 ATP molecules.

**Figure 2 nutrients-13-04433-f002:**
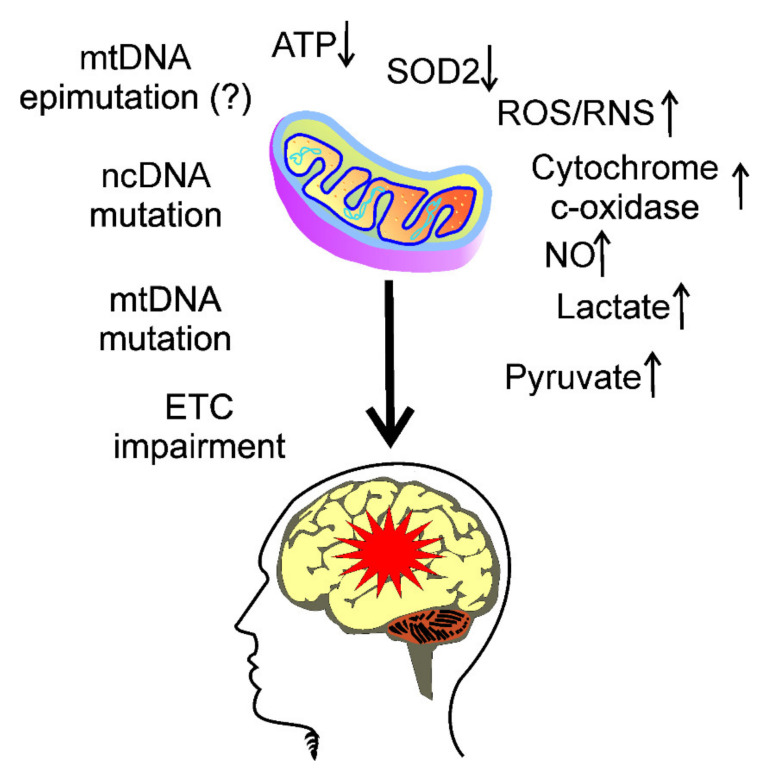
Migraine associates with mitochondrial dysfunction and energy production. Impairment of the electron transport chain (ETC) in mitochondria may result in decreased ATP production. There are several reasons and consequences of this impairment, including mutations and epimutations in both mitochondrial DNA (mtDNA) and nuclear DNA (ncDNA), decrease in the activity of mitochondrial superoxide dismutase (SOD2), which, along with impaired ETC, may contribute to increased production of reactive oxygen and nitrogen species (ROS and RNS, respectively), activation of cytochrome-c oxidase and nitric oxide, high levels of lactate and pyruvate, as well as other effects, which together may be involved in migraine headache (red star) induction.

**Figure 3 nutrients-13-04433-f003:**
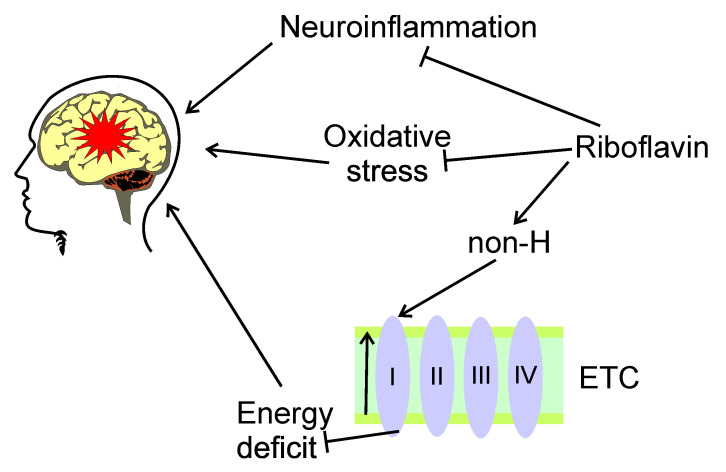
Potential beneficial effects of riboflavin in migraine. Riboflavin may exert protective effects in migraine, symbolized here by a red star, by the modulation of the mitochondria-energy production-oxidative stress pathway, but its details are poorly known. Riboflavin may reduce oxidative stress associated with migraine. Riboflavin targets complex I of the electron transport chain (ETC, simplified here to complexes I–IV), whose activity may decrease in migraineurs, especially those with non-H mitochondrial haplotype (non-H). Riboflavin may ameliorate energy deficit resulted from malfunction of the complex I, and, in this way, decreases migraine-related symptoms. These activities result from the involvement of riboflavin in mitochondrial homeostasis. Riboflavin may also reduce neuroinflammation, important in migraine pathogenesis.

**Figure 4 nutrients-13-04433-f004:**
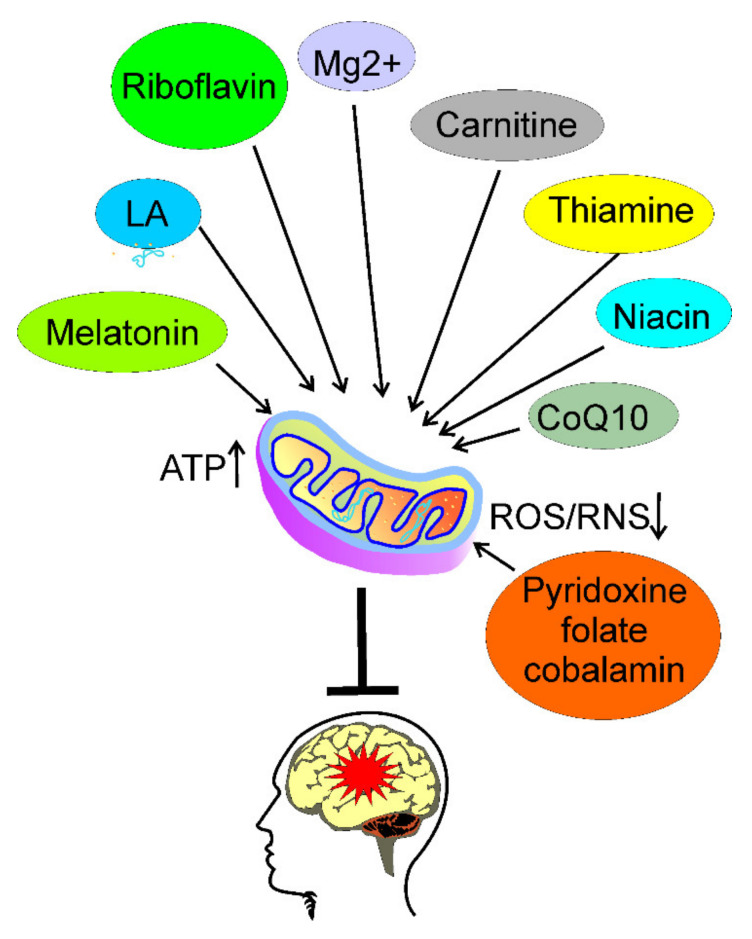
Some nutrients may prevent or cure migraine. Certain nutrients may improve energy production, and reduce energy deficit in the brain through improvement of mitochondrial functions, resulting in increased ATP production, and reduction of oxidative stress through decreasing reactive oxygen and nitrogen species (ROS and RNS, respectively). Such an action may prevent or ameliorate migraine (red star). Some candidates for such compounds are: riboflavin; thiamine; magnesium ions (Mg2+); niacin; carnitine; coenzyme Q10 CoQ10); melatonin; lipoic acid (LA); pyridoxine; folate; and cobalamin.

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
