# Peer review of "Nutrients to Improve Mitochondrial Function to Reduce Brain Energy Deficit and Oxidative Stress in Migraine"

_nutrients, 2021, doi:10.3390/nu13124433_

Round 1

Reviewer 1 Report

Journal: Nutrients

Article type: Review.

Authors: Fila et al.

Title: Nutrients to improve mitochondrial function to reduce brain energy deficit and oxidative stress in migraine.

This is a well-planned, interesting review, with potential relevance in clinical practice.

The Authors discuss the possible role of mitochondrial metabolism/dysfunction in migraine pathogenesis and the link between several nutrients, that are important for mitochondrial function, brain energy balance and modulation of cellular oxidative stress, wth the severity and frequency of migraine attacks.

Major criticisms and suggestions:

- Among the myriad of nutrients potentially involved, the Authors should also discuss the possible role of pyridoxine, a B vitamin with definite antioxidant properties, folate and cobalamin supplementation in the pathophysiology and treatment of migraine attacks.

Moreover, the relationship between serum levels of homocysteine, the remethylation and folate cycles (to see, Sechi et al., Nutrition Reviews 2016), with the frequency and characteristics of attacks in patients with migraine should be further discussed. (Sadeghi et al., J Res Med Sci, 2014; Askari et al., Nutrition, 2017; Sadeghi et al., Iran J Neurol, 2015; Menon et al., J Headache Pain, 2016; Lippi et al., Clin Chim Acta, 2014). The above articles should be quoted in the Review.

- Page 11, line 1: Coenzyme Q10 has been used at the dose of 150 mg per day (not 6 mg !). Please, correct this inaccuracy and check the entire manuscript for possible similar errors.

Minor criticisms:

- Figure1. Puryvate should be corrected in Pyruvate.

- Figure 4 should be modified according to the suggested revisions.

Author Response

This is a well-planned, interesting review, with potential relevance in clinical practice.

The Authors discuss the possible role of mitochondrial metabolism/dysfunction in migraine pathogenesis and the link between several nutrients, that are important for mitochondrial function, brain energy balance and modulation of cellular oxidative stress, wth the severity and frequency of migraine attacks.

Major criticisms and suggestions:

Comment: - Among the myriad of nutrients potentially involved, the Authors should also discuss the possible role of pyridoxine, a B vitamin with definite antioxidant properties, folate and cobalamin supplementation in the pathophysiology and treatment of migraine attacks.

Answer: As pyridoxine, folate and cobalamin lower homocysteine (Hcy) level and hyperhomocysteinemia may be important in migraine pathogenesis, we described the role of these compound collectively in a single sub-section 5.9 Pyridoxine, folate and cobalamin to keep the review more concise.  

5.9 Pyridoxine, folate and cobalamin

Migraine was associated with vascular comorbidities, including stroke [147]. Lowering the levels of homocysteine (Hcy) was reported to exert protective effects against detrimental vascular events that were often characterized by hyperhomocysteinemia (reviewed in [148,149]). Several studies showed a correlation between serum Hcy levels and frequency and characteristics of migraine attacks (reviewed in [150]). Moreover, the mechanism of the involvement of Hcy in vascular disorders, including stroke, may be related to folate cycles and remethylation of Hcy to methionine in the central nervous system (reviewed in [151]). We previously suggested that these effects might play a role in migraine pathogenesis [152]. Elevated levels of Hcy were associated with disturbances in the functioning of mitochondria and energy production in the central nervous system (reviewed in [153]). Sadeghi et al. did not find any correlation between serum Hcy levels and characteristics of migraine, including severity, duration, frequency and HDR in a cross-sectional study with 120 patients [154]. However, such a correlation between serum Hcy levels and HDR was observed in male patients after adjustment for age, BMI and family history of migraine.

Dietary compounds, pyridoxine (vitamin B6), folic acid and cobalamin (vitamin B12) were shown to lower Hcy levels in many pathologies [151,155-157]. Moreover, these compounds are also involved in mitochondrial homeostasis, energy production and antioxidant defense in the central nervous system [158-160].

Several studies suggest a beneficiary effect of pyridoxine, folate and cobalamin supplementation in migraineurs. Sadeghi et al. observed that pyridoxine intake (80 mg/day) in 66 patients with migraine with aura resulted in beneficiary effects on headache severity, HDR and migraine attacks duration, but not their frequency [161]. A significant decrease in migraine severity, frequency and duration of migraine attacks and HDR in 95 patients with migraine with aura with the diet supplementation with pyridoxine (80 mg/day) and folic acid (5 mg/day) for 3 months as compared with placebo or folic acid only [162]. Lea et al. administrated for 6 months 2 mg of folic acid, 25 mg pyridoxine, and 400 µg cobalamin daily to 52 patients with migraine with aura [163]. In that study, a moderate effect of the genotypes of the c.677C>T polymorphism in the MTHFR (methylenetetrahydrofolate reductase) gene was also observed. This polymorphism was reported to associate with migraine susceptibility in numerous studies (reviewed in [164]). The same group observed in a subsequent study that lowering the dose of folic acid to 1 mg/day resulted in a less effective reducing migraine syndromes after the combined folic acid/pyridoxine/cobalamin supplementation [165]. A case-control study on 70 migraine patients and 70 individuals without headache showed that individuals with a lower status of cobalamin and methylmalonic acid had higher odds for migraine [166].

In summary, many studies suggest that supplementation with folic acid, pyridoxine and cobalamin may exert a beneficial effect in migraine, first in adults with aura. Therefore, further studies with other age groups are required. Moreover, such dietary intervention should be studied with genotyping of the c.677C>T polymorphism in the MTHFR gene as many studies suggest its involvement in migraine pathogenesis.

with new references:

  1. Buse, D.C.; Reed, M.L.; Fanning, K.M.; Bostic, R.; Dodick, D.W.; Schwedt, T.J.; Munjal, S.; Singh, P.; Lipton, R.B. Comorbid and co-occurring conditions in migraine and associated risk of increasing headache pain intensity and headache frequency: results of the migraine in America symptoms and treatment (MAST) study. The journal of headache and pain 2020, 21, 23, doi:10.1186/s10194-020-1084-y.
  2. Jin, N.; Huang, L.; Hong, J.; Zhao, X.; Chen, Y.; Hu, J.; Cong, X.; Xie, Y.; Pu, J. Elevated homocysteine levels in patients with heart failure: A systematic review and meta-analysis. Medicine 2021, 100, e26875, doi:10.1097/md.0000000000026875.
  3. Smith, A.D.; Refsum, H. Homocysteine - from disease biomarker to disease prevention. J Intern Med 2021, 290, 826-854, doi:10.1111/joim.13279.
  4. Lippi, G.; Mattiuzzi, C.; Meschi, T.; Cervellin, G.; Borghi, L. Homocysteine and migraine. A narrative review. Clin Chim Acta 2014, 433, 5-11, doi:10.1016/j.cca.2014.02.028.
  5. Sechi, G.; Sechi, E.; Fois, C.; Kumar, N. Advances in clinical determinants and neurological manifestations of B vitamin deficiency in adults. Nutr Rev 2016, 74, 281-300, doi:10.1093/nutrit/nuv107.
  6. Fila, M.; Chojnacki, C.; Chojnacki, J.; Blasiak, J. Is an "Epigenetic Diet" for Migraines Justified? The Case of Folate and DNA Methylation. Nutrients 2019, 11, doi:10.3390/nu11112763.
  7. Kaplan, P.; Tatarkova, Z.; Sivonova, M.K.; Racay, P.; Lehotsky, J. Homocysteine and Mitochondria in Cardiovascular and Cerebrovascular Systems. International journal of molecular sciences 2020, 21, doi:10.3390/ijms21207698.
  8. Sadeghi, O.; Maghsoudi, Z.; Askari, G.; Khorvash, F.; Feizi, A. Association between serum levels of homocysteine with characteristics of migraine attacks in migraine with aura. Journal of research in medical sciences : the official journal of Isfahan University of Medical Sciences 2014, 19, 1041-1045.
  9. Christen, W.G.; Cook, N.R.; Van Denburgh, M.; Zaharris, E.; Albert, C.M.; Manson, J.E. Effect of Combined Treatment With Folic Acid, Vitamin B(6), and Vitamin B(12) on Plasma Biomarkers of Inflammation and Endothelial Dysfunction in Women. J Am Heart Assoc 2018, 7, doi:10.1161/jaha.117.008517.
  10. Dusitanond, P.; Eikelboom, J.W.; Hankey, G.J.; Thom, J.; Gilmore, G.; Loh, K.; Yi, Q.; Klijn, C.J.; Langton, P.; van Bockxmeer, F.M.; et al. Homocysteine-lowering treatment with folic acid, cobalamin, and pyridoxine does not reduce blood markers of inflammation, endothelial dysfunction, or hypercoagulability in patients with previous transient ischemic attack or stroke: a randomized substudy of the VITATOPS trial. Stroke 2005, 36, 144-146, doi:10.1161/01.Str.0000150494.91762.70.
  11. Hankey, G.J.; Eikelboom, J.W.; Loh, K.; Tang, M.; Pizzi, J.; Thom, J.; Yi, Q. Sustained homocysteine-lowering effect over time of folic acid-based multivitamin therapy in stroke patients despite increasing folate status in the population. Cerebrovasc Dis 2005, 19, 110-116, doi:10.1159/000082788.
  12. Calderón-Ospina, C.A.; Nava-Mesa, M.O. B Vitamins in the nervous system: Current knowledge of the biochemical modes of action and synergies of thiamine, pyridoxine, and cobalamin. CNS Neurosci Ther 2020, 26, 5-13, doi:10.1111/cns.13207.
  13. Depeint, F.; Bruce, W.R.; Shangari, N.; Mehta, R.; O'Brien, P.J. Mitochondrial function and toxicity: role of B vitamins on the one-carbon transfer pathways. Chemico-biological interactions 2006, 163, 113-132, doi:10.1016/j.cbi.2006.05.010.
  14. Maggini, S.; Pierre, A.; Calder, P.C. Immune Function and Micronutrient Requirements Change over the Life Course. Nutrients 2018, 10, doi:10.3390/nu10101531.
  15. Sadeghi, O.; Nasiri, M.; Maghsoudi, Z.; Pahlavani, N.; Rezaie, M.; Askari, G. Effects of pyridoxine supplementation on severity, frequency and duration of migraine attacks in migraine patients with aura: A double-blind randomized clinical trial study in Iran. Iran J Neurol 2015, 14, 74-80.
  16. Askari, G.; Nasiri, M.; Mozaffari-Khosravi, H.; Rezaie, M.; Bagheri-Bidakhavidi, M.; Sadeghi, O. The effects of folic acid and pyridoxine supplementation on characteristics of migraine attacks in migraine patients with aura: A double-blind, randomized placebo-controlled, clinical trial. Nutrition (Burbank, Los Angeles County, Calif.) 2017, 38, 74-79, doi:10.1016/j.nut.2017.01.007.
  17. Lea, R.; Colson, N.; Quinlan, S.; Macmillan, J.; Griffiths, L. The effects of vitamin supplementation and MTHFR (C677T) genotype on homocysteine-lowering and migraine disability. Pharmacogenet Genomics 2009, 19, 422-428, doi:10.1097/FPC.0b013e32832af5a3.
  18. Liu, L.; Yu, Y.; He, J.; Guo, L.; Li, H.; Teng, J. Effects of MTHFR C677T and A1298C Polymorphisms on Migraine Susceptibility: A Meta-Analysis of 26 Studies. Headache 2019, 59, 891-905, doi:10.1111/head.13540.
  19. Menon, S.; Nasir, B.; Avgan, N.; Ghassabian, S.; Oliver, C.; Lea, R.; Smith, M.; Griffiths, L. The effect of 1 mg folic acid supplementation on clinical outcomes in female migraine with aura patients. The journal of headache and pain 2016, 17, 60, doi:10.1186/s10194-016-0652-7.
  20. Togha, M.; Razeghi Jahromi, S.; Ghorbani, Z.; Martami, F.; Seifishahpar, M. Serum Vitamin B12 and Methylmalonic Acid Status in Migraineurs: A Case-Control Study. Headache 2019, 59, 1492-1503, doi:10.1111/head.13618.

Comment: Moreover, the relationship between serum levels of homocysteine, the remethylation and folate cycles (to see, Sechi et al., Nutrition Reviews 2016), with the frequency and characteristics of attacks in patients with migraine should be further discussed. (Sadeghi et al., J Res Med Sci, 2014; Askari et al., Nutrition, 2017; Sadeghi et al., Iran J Neurol, 2015; Menon et al., J Headache Pain, 2016; Lippi et al., Clin Chim Acta, 2014). The above articles should be quoted in the Review.

Answer: We shortly described the relationship between homocysteine and migraine as an introductory paragraph to the new 5.9 section.

Comment:- Page 11, line 1: Coenzyme Q10 has been used at the dose of 150 mg per day (not 6 mg !). Please, correct this inaccuracy and check the entire manuscript for possible similar errors.

Answer: We have corrected that mistake.

Minor criticisms:

Comment: Figure1. Puryvate should be corrected in Pyruvate.

Answer: We have corrected that.

Comment: - Figure 4 should be modified according to the suggested revisions.

Answer: We have corrected Figure 4 accordingly.

Reviewer 2 Report

The review work presented by Fila and colleagues is well written, clear, and easy to read. The topic is interesting and therefore, it adds clustered information to the subject area of migraine compared with others published articles. In particular, the author points out the pros and cons of different kinds of supplements used in the management of migraine onset. For me it can be accepted with minor revision concerning the part of the diet, they did not list at all the ketogenic diet. Please consider these two papers which are opening new options for migraine patients.

https://doi.org/10.3390/jcm8070928

doi: 10.1093/cdn/nzz052.P14-007-19

Author Response

The review work presented by Fila and colleagues is well written, clear, and easy to read. The topic is interesting and therefore, it adds clustered information to the subject area of migraine compared with others published articles. In particular, the author points out the pros and cons of different kinds of supplements used in the management of migraine onset.

Comment: For me it can be accepted with minor revision concerning the part of the diet, they did not list at all the ketogenic diet. Please consider these two papers which are opening new options for migraine patients.

https://doi.org/10.3390/jcm8070928

doi: 10.1093/cdn/nzz052.P14-007-19

Answer: Firstly, as we state in our manuscript, we are somehow sceptical to many “specific” diets that are claimed to exert a beneficiary effect in some diseases, including migraine. In our manuscript we have formulated our view on how a migraine-oriented diet should be. Therefore, we do not recommend the keto diet in the context of migraine, energy production and mitochondria amelioration. The first recommended paper is an abstract of apparently local meeting (comma instead of full stops as decimal point) describing 6 obese migraineurs with keto diet, but in the context of miRNA regulation. We do not consider is as a breakthrough in the field and therefore we do not include it in our manuscript. The second paper is on miRNA regulation to monitor drug action in children and adolescent suffering from migraine without aura. We were not able to find any connotation with diet in that manuscript. Although some of these miRNAs in both papers are linked with metabolism, this is another, for sure important, subject deserving for a separate review.